# Two steps to risk sensitivity

**Chris Gagne**
MPI for Biological Cybernetics

Tübingen, Germany
christopher.gagne@tuebingen.mpg.de

**Peter Dayan**
MPI for Biological Cybernetics
University of Tübingen
Tübingen, Germany
dayan@tue.mpg.de

## Abstract

Distributional reinforcement learning (RL) – in which agents learn about all the possible long-term consequences of their actions, and not just the expected value – is of great recent interest. One of the most important affordances of a distributional view is facilitating a modern, measured, approach to risk when outcomes are not completely certain. By contrast, psychological and neuroscientific investigations into decision making under risk have utilized a variety of more venerable theoretical models such as prospect theory that lack axiomatically desirable properties such as coherence. Here, we consider a particularly relevant risk measure for modeling human and animal planning, called conditional value-at-risk (CVaR), which quantifies worst-case outcomes (e.g., vehicle accidents or predation). We first adopt a conventional distributional approach to CVaR in a sequential setting and reanalyze the choices of human decision-makers in the well-known two-step task, revealing substantial risk aversion that had been lurking under stickiness and perseveration. We then consider a further critical property of risk sensitivity, namely time consistency, showing alternatives to this form of CVaR that enjoy this desirable characteristic. We use simulations to examine settings in which the various forms differ in ways that have implications for human and animal planning and behavior.

## 1 Introduction

Risk is integral to decision making, arising whenever there are uncertain outcomes. It is especially critical when those outcomes are potentially calamitous, and plays an important role in psychiatric illness [1]. However, psychological investigations into choice under risk (i) have yet to embrace the strong formal foundations being developed in finance, AI and machine learning; and (ii) have mostly focused on one-shot decision tasks, despite the ubiquity of situations in the real-world that require planning, and growing interest in multi-step tasks for elucidating richer mechanisms of choice [2, 3].

To address these lacunæ, we consider a modern, theoretically well-founded coherent [4] risk measure called conditional value-at-risk (CVaR). This exactly quantifies the lower tail of possible outcomes – those which are important for survival and also tend to drive our most persistent worries. CVaR has been applied to sequential decision problems, notably by means of distributional reinforcement learning (RL; [5–7]). In this paradigm, the agent (or decision maker) estimates whole distributions for potential outcomes that can arise from its actions. Risk measures, such as CVaR, can be applied to these distributions to adjust decision making to any desired level of risk sensitivity.

In this paper, we combine CVaR with a distributional approach to examine risk sensitivity in the intensively-investigated two-step sequential decision task [2]. We find that the choices of many individuals reflect a large degree of risk aversion. Moreover, we show that more standard analyses ascribe this to enhanced choice perseveration and/or reduced estimates for learning rate, thus misspecifying the effects of relatively high levels of uncertainty. Incorporating risk sensitivity into

35th Conference on Neural Information Processing Systems (NeurIPS 2021).

choice, through CVaR or other risk measures, however, raises subtle issues regarding the consistency of choices over time [8–11]. Such issues are well explored in psychology in the context of hyperbolic temporal discounting [12], but their thorough investigation in finance and operations research has yet to permeate judgement and decision making research. We discuss how a direct incorporation of CVaR into distributional RL can lead to time inconsistency and point to two additional time-consistent approaches to CVaR for sequential decision making. Finally, we show how these three approaches can give rise to distinct patterns of behavior in a problem designed to tease them apart and which can serve as the basis for future empirical investigation.

**Preliminaries: Conditional value-at-risk (CVaR)**

A risk measure $\rho(Z)$ maps a probability distribution of outcomes associated with a random variable $Z$ to a real number. In distributional RL, $Z$ is typically a discounted sum of rewards minus costs (i.e. the return). $\rho(Z)$ represents the risk inherent in the uncertainty about $Z$, and is often interpreted as the amount one is willing to pay to avoid adopting this risk.

Well known risk measures include the variance and the value-at-risk ($\text{VaR}_\alpha$), which is defined as:

$$\text{VaR}_\alpha(Z) = F^{-1}(\alpha) := \inf\{z : F(z) \geq \alpha\} \tag{1}$$

for a cumulative distribution function $F(z)$ and $\alpha$-quantile; see Figure 1a. While variance and VaR are venerable risk measures, neither satisfy the full set of axiomatically desirable properties associated with *coherent risk measures* [4]. For instance, the VaR is not sub-additive (so fails to reward diversification), and variance is neither sub-additive nor positively homogeneous (as it changes non-linearly with the units in which costs or rewards are measured).

Conditional value-at-risk (CVaR) was introduced as a coherent risk measure [4], and is particularly popular in finance, robotics, operations research, and recently machine learning and RL [6, 13–19]. For the lower tail of a continuous distribution, it is defined as the average of the values lower than the VaR (Figure 1a):[1]

$$\text{CVaR}_\alpha[Z] := E[Z|Z < \text{VaR}_\alpha(Z)] \tag{2}$$

$\alpha$ determines risk aversion by emphasizing the lower tail of the distribution more or less (Figure 1b).

That CVaR concentrates on the lower tail makes it particularly attractive for capturing aspects of normal and pathological reasoning in animals, whose lives often hang on rather thin threads, and humans, who can catastrophize about diverse, unlikely, possibilities. But it is then particularly important to consider CVaR in sequential decision making, since this represents the ecological norm and is etched into the neural structure of decision making. We therefore start by examining CVaR in the simplest such problem that is widely studied in humans: the two-step task.

## 2 Modeling risk sensitivity in human planning: CVaR in the two-step task

The two-step task (Figure 2) is a popular option for studying sequential choice in humans (and animals) [2]. It was originally designed to investigate model-based (MB) and model-free (MF) learning and planning, distinguishing the two by examining the consequences of progressive changes in the probabilities of rewards. However, these changes necessarily induce uncertainty – which could affect risk-sensitive subjects. We therefore used a CVaR-based form of MB and MF reasoning to fit a very substantial dataset of human behavior in this task (out of more than 2000 participants in [3], the 792 who responded on every trial).

**Task:** On each of 200 trials, participants make decisions at two successive stages or steps. At the first stage, participants choose between two actions (the fractals in Figure 2a). As shown by the arrows in the figure, depending on their choice, they then transition to one of two possible second stage states either 70% or 30% of the time. Each second stage state has its own pair of options (the colored squares) between which the participants must then choose. According to this second choice, the participant receives a binary outcome, which is drawn according to a probability that drifts randomly across the trials (shown by color in Figure 2b). Before the actual experiment, participants were given 40 trials of practice, during which they learned about the general structure of the task

---

[1]For discrete random variables, alternative definitions such as $\text{CVaR}_\alpha(Z) := \sup_\nu\{\nu - \frac{1}{\alpha}E[(\nu - Z)^+]\}$ [20] are used.

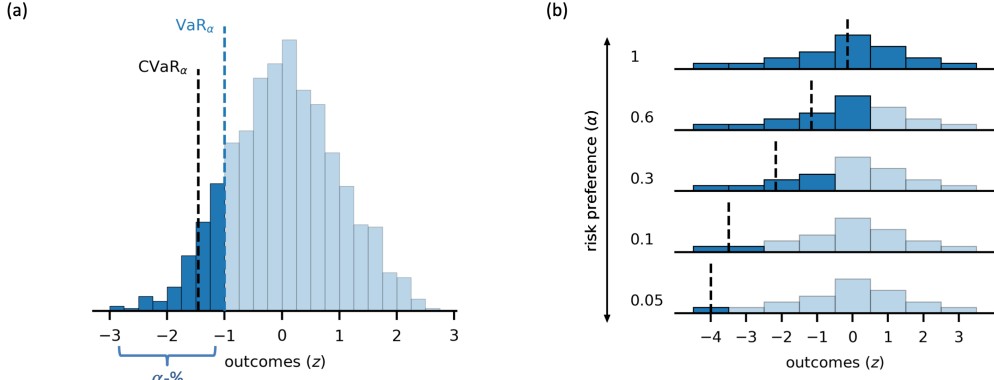

Figure 1: Conditional value-at-risk (CVaR). (a) CVaR$_\alpha$ is the average of the values in the lower $\alpha$-% of a distribution, i.e. below the VaR$_\alpha$. (b) Adjusting $\alpha$ emphasizes the lower tail of the distribution more or less and sets the level of risk aversion. At $\alpha = 1.0$, CVaR is equal to the mean. As $\alpha$ decreases to $0$, it approaches the minimum.

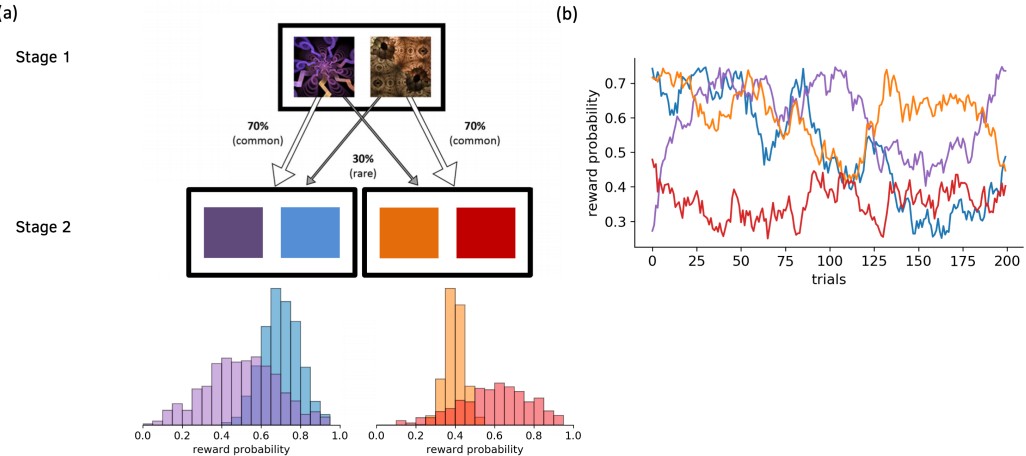

Figure 2: Two-step task. (a) On each trial, participants choose between two options (depicted as fractals) in stage 1 and then transition stochastically to one of two states (purple/blue or orange/red) in stage 2. They again choose between two options (the colored squares) and receive a binary reward. (b) This reward is sampled according to a probability that drifts randomly across trials (separately for each option). Distributions over reward probabilities are assumed by the model to be estimated for each option (depicted at the bottom in panel a).

including the 70%/30% transition probabilities. This structure was not presented to participants explicitly, but participants were tested about their knowledge of the task and excluded if they failed this test. Participants were instructed about the drifting reward probabilities.

**CVaR-based model:** We included CVaR in a model that is conventionally applied to the two-step task. We assume that participants learn a distribution of values [5] associated with each of the four second-stage options using an approximate Kalman filter:

$$\mu_{t+1} = \mu_t + \lambda(r_t - \mu_t) \qquad \Psi_{t+1}^2 = (1 - \phi^2)\Psi_t^2 + \eta^2 - \lambda\Psi_t^2 \qquad (3)$$

updating the mean $\mu_t$ of each distribution on each trial using the observed outcome $r_t$ via a delta-rule with a participant-specific learning rate $\lambda$, and also updating the variance $\Psi_t$. When the outcome is not observed (i.e. when the participant chose a different option), the mean is updated towards 0.5 and the variance is updated without the last term in equation (3). Thus, the dispersion parameter $\eta^2$ controls the increase of the variance whether or not outcomes are observed, and the learning rate $\lambda$ controls the amount it decreases when they are. The term $(1 - \phi^2)$ controls the asymptotic variance for unobserved outcomes (in relation to $\eta^2$) and was set based on the other two parameters (see later).

The $\text{CVaR}_{\alpha,t}(a)$ at risk preference $\alpha$ (estimated per participant) was calculated for each option $a$ using the mean $\mu_t(a)$ and variance $\Psi_t(a)$ on each trial $t$. Note that because this variance represents uncertainty about the reward probabilities themselves, CVaR here captures both ambiguity- as well as risk-sensitivity; however, to keep terminology consistent with the finance and machine learning communities [21–24], we simply refer to this as to risk-sensitivity.

Equipped with these $\text{CVaR}_{\alpha,t}(a)$ values, participants' second stage choices were modeled using a soft-max choice rule:

$$P(2^{nd} \text{ stage choice} = a) \propto \exp(\beta^{2nd}\text{CVaR}_{\alpha,t}(a)) \tag{4}$$

where the parameter $\beta^{2nd}$ controls the relative stochasticity/determinism.

Decisions between the two first-stage options were modeled as involving a combination of model-free (MF) and model-based (MB) approaches to value estimation – both of which were modified to include CVaR. MF estimates were calculated using the same formulæ, learning rate $\lambda$ and dispersion parameter $\eta^2$ as at the second stage to learn an additional pair of means and variances based on the actual outcomes received in the second stage. For model-based estimation, the 70%/30% transition probabilities were used to calculate a mixture distribution for each of the top stage actions, from which the CVaR was calculated directly. $\text{CVaR}_{\alpha,t}(a)$ for the model-free and model-based first-stage distributions again determined the choice probability through a soft-max choice rule:

$$P(1^{st}\text{stage choice} = a) \propto \exp(\beta^{MB}\text{CVaR}^{MB}_{\alpha,t}(a) + \beta^{MF}\text{CVaR}^{MF}_{\alpha,t}(a) + \beta^{\text{sticky}}\delta_{a,a_{t-1}}) \tag{5}$$

Parameter $\beta^{\text{sticky}}$ was included to capture the tendency of participants to repeat (or to switch) the previously chosen action regardless of its value.

**Parameter estimation:** The 7 parameters of the CVaR-model were estimated for each participant: CVaR-based risk-sensitivity $\alpha \in [0.1, 1]$, learning rate $\lambda \in [0.01, 0.99]$, dispersion $\eta^2 \in [0.001, 0.09]$, perseveration $\beta^{\text{sticky}} \in [0, 20]$ and three inverse temperature parameters $\beta^{2nd}, \beta^{MB}, \beta^{MF} \in [0, 30]$. Parameters were estimated in Python using L-BFGS-B. Parameter recovery analyses were conducted to investigate parameter estimability. Preliminary recovery simulations suggested that estimating both $\eta^2$ and $(1 - \phi^2)$ was difficult, so we determined the value of $(1 - \phi^2)$ that would pin the asymptotic variance to 0.1. This was done so that a never-chosen option with an estimated mean 0.5 would have a CVaR of 0 at the lower-bound value for $\alpha$ (i.e. 0.1), which is appropriate since the outcomes themselves were between 0 and 1. The learning rate lay between 0.01 to 0.99 and was additionally constrained such that $\lambda < (1 - \phi^2)$. With these constraints, we ran further parameter recovery analyses, generating new data based on participants' parameter estimates and re-estimating the model; a rank-correlation of 0.71 between the generative and recovered CVaR $\alpha$ parameter indicated moderately good estimability/identifiability; Supplemental Figure 1.

**Baseline and alternate models:** Setting $\alpha = 1$ arranges for CVaR choices that depend only on expected values (in which case the variance estimation can also be removed, because it does not influence choice probability). This mean-model is used as a baseline against the risk-sensitive $\text{CVaR}_{\alpha<1}$ model; it is very similar to typical models used in the two-step task [2, 3], but is more directly comparable to its risk-sensitive counterpart. We provide a more detailed description of the differences between the current and previously used model in Appendix A.2. We also tried two forgetful beta-binomial models (one with and without risk-sensitivity) since rewards are binary, and a risk-seeking version of the CVaR-model, which emphasized the upper rather than lower tail. However, these models fit participants' data less well, so we omit them for the sake of parsimony.

## Results

We first compared the the mean-model ($\text{CVaR}_{\alpha=1}$) to the risk-sensitive model ($\text{CVaR}_{\alpha<1}$). Using BIC to account for the two extra parameters, the risk-sensitive model was slightly preferred overall (avg. BIC=368.8 versus BIC=371.5; diff=-2.67, t(790)=-3.54, p=0.004), albeit in a minority (40.2%) of participants. For many such subjects, however, the improved fit was substantial (red points in Figure 3a); and many of them had $\alpha < 0.2$ (Figure 3b), indicating substantial aversion to risk.

To investigate the improved fit and for insight into the choice characteristics associated with this risk aversion, we analyzed how other parameters altered from the $\alpha = 1$ case for significantly risk-averse

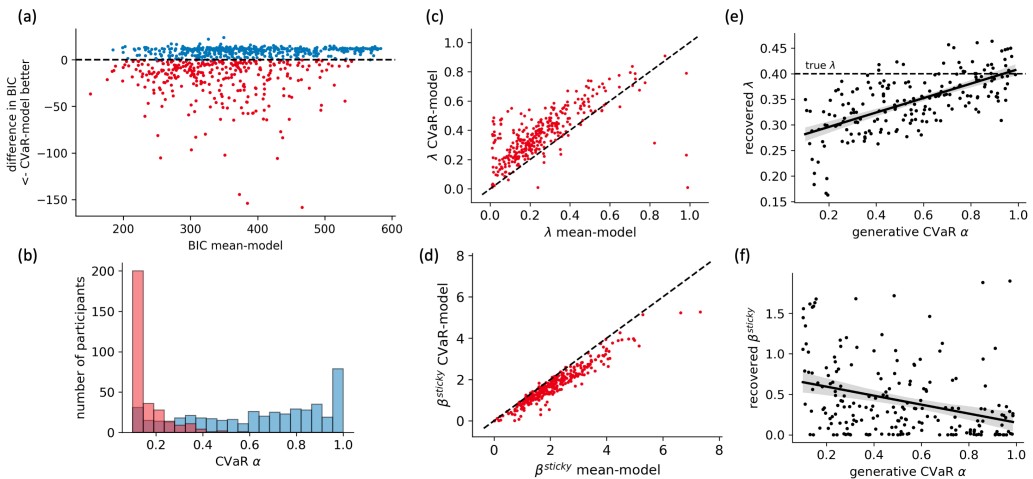

Figure 3: Two-step task modeling results. (a) Many participants are better fit by the CVaR-model than the mean-model, and (b) are substantially risk averse, as estimated by the CVaR $\alpha$ parameter. (c-d) Including risk sensitivity increases/decreases estimates of learning rate and perseveration, respectively, in these participants. (e-f) Simulations show that fitting a risk-agnostic model to risk-sensitive behavior can lead to underestimated learning rates (e) and/or apparent perseveration (f). Shading around the regression line in panels e-f indicate 95%-confidence intervals. Note that 3 outliers located near (x=12, y=10) were removed from panel d.

subjects. There were two related changes: participants' estimated learning rates were higher in the CVaR-model (Figure 3c), and their estimated perseveration parameters were lower (Figure 3d). One potential reason for these is that participants tended *not* to prefer less frequently chosen (and thus more uncertain) options, even when the more certain option is apparently worse.

To demonstrate this directly, we simulated choices from the CVaR-model at increasing levels of risk sensitivity ($\alpha \in [1.0, 0.6, 0.3, 0.1]$) in response to a predetermined set of outcomes, with the other parameters held constant (Figure 4). Option A is chosen by design for the first 6 trials, but the model is then allowed to switch to an option B after observing what it thinks is a sufficient number of negative outcomes. For $\alpha = 1$, this switch occurs when the mean estimate for option A crosses the mean estimate for option B at 0.5 (trial 10 in panel b). For $\alpha < 1$ (panels c-e), this switch occurs when the CVaR$_{\alpha<1.0}$ estimate for the options cross. Due to the uncertainty around the mean and the fact that the model is more uncertain about option B than A, the crossing point occurs later for lower $\alpha$ (i.e. on trials 11, 12, and 13). Thus, what looks like a tendency to stick with an apparently worse option, here arises from an aversion to uncertainty. For $\alpha = 1$, this can only occur if the learning rate is low (so the value of the chosen option does not decrease too much upon experiencing an unfortunate outcome), and/or with a high perseveration parameter. But, as a consequence, genuine risk aversion (or uncertainty aversion) might be misattributed to one of these parameters – this is shown directly in Figure 3e;f where generating choices from a non-perseverative CVaR-model with low $\alpha$ leads to inferred learning rates that are too low (panel e) and inferred non-zero perseveration (panel f). Note, though, that in participants' data, some perseveration remains even with CVaR$_{\alpha<1}$, as removing perseveration from the CVaR-model (i.e. setting $\beta^{\text{sticky}} = 0$) led to worse BIC values.

## 3   Three approaches to sequential risk and time (in)consistency

One of the most important issues for sequential decision making is time consistency – that the choices the decision maker at time $t$ assumes will be executed at time $t + 1$ are indeed carried out.[2] Inconsistency, famously caused by hyperbolic discounting [12], can lead to reneging on past resolutions or require potentially sub-optimal and expensive commitment behavior to prevent that

---

[2]An intimately related, yet differently defined, notion of time consistency involves the consistency of successive risk evaluations of a stochastic process rather than choices per se. Of course, inconsistent evaluation can lead to inconsistent choices. See [25–27] for a more in-depth discussion.

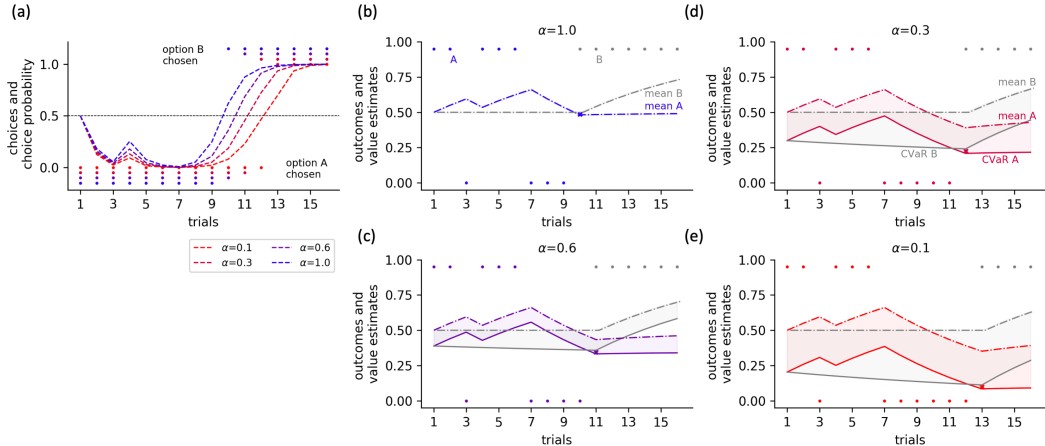

Figure 4: CVaR-based perseveration. (a) The CVaR-model is simulated at four values of $\alpha$ on a series of choices between options A and B. For low $\alpha$ (high risk sensitivity), the model chooses option A for more trials, despite receiving 0 outcomes after trial 7; binary outcomes are shown as dots in panels b-e. (b-e) The switch from A to B occurs when the model's estimate for the CVaR (or mean for $\alpha = 1.0$) of A (in color) goes below that of B (in gray). The crossing point occurs later for lower levels of $\alpha$ due to uncertainty around the mean which is greater for B.

from occurring. Another, less well-known, route to time inconsistency comes from mishandling risk sensitivity [8–11, 26–34] – and, indeed, different variants of CVaR have been developed to deal with this issue. The two-step task lacks sufficient temporal complexity to make this issue clear. Therefore in this section we examine it in greater depth, and, using simulations, suggest a task that highlights the differences between these CVaR variants.

**Fixed, precommitted, and nested CVaR**

In the two-step task, we followed recent work on distributional RL [6] in assuming that participants applied CVaR at the same risk preference ($\alpha$) to the estimated value distributions at both the first and

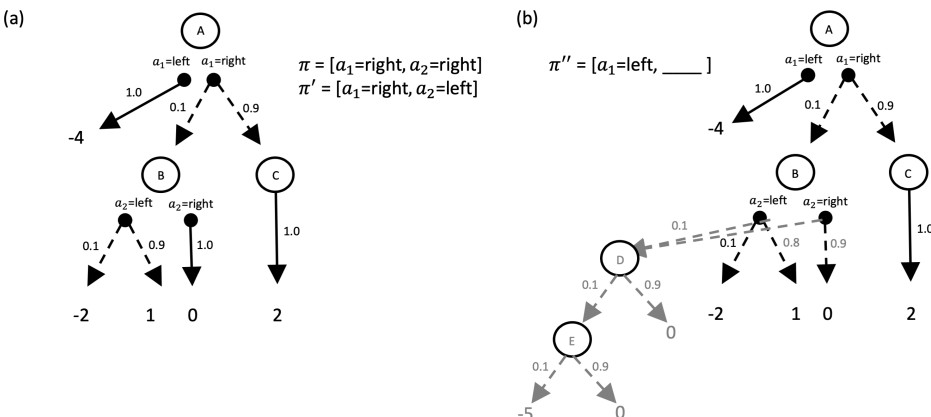

Figure 5: Three approaches to CVaR in a two-stage problem. (a) A choice of left or right can be made in both state A and B (if state B is visited). Stochastic transitions are denoted by dashed lines (with the probabilities shown), and the possible terminal outcomes are [-4, -2, 1, 0, 2]. The fixed approach to CVaR (discussed in the text) would choose policy $\pi'$ at state A, but policy $\pi$ at state B, illustrating issues with time consistency. The precommitted approach would consistently choose policy $\pi'$. (b) In an adjusted problem, additional state transitions (colored in gray) are appended to state B. The nested approach to CVaR (with $\alpha = 0.1$) chooses policy $\pi''$, taking a sure loss of $-4$, despite the only extremely remote chance of getting $-5$.

second stage. We call this the fixed approach or fCVaR. To see why this approach is time-*in*consistent, consider calculating the optimal fCVaR$_{\alpha=0.1}$ policy for the simplified two-stage choice scenario depicted in Figure 5a.

From the perspective of the top state A, consider the two policies $\pi = \{a_1 = right, a_2 = right\}$, which leads to a distribution of outcomes $p = \{0.1 : 0, 0.9 : 2\}$ and $\pi' = \{a_1 = right, a_2 = left\}$, which leads to a distribution of outcomes $p' = \{0.01 : -2, 0.09 : 1, 0.9 : 2\}$. Just considering these overall distributions, $\pi$ has a CVaR$_{0.1}$ of 0 and $\pi'$ has an CVaR$_{0.1}$ of 0.7, so $\pi'$ would be preferred. However, if the agent takes action $a_1 = right$ and arrives at state B, the remainder of $\pi'$ (i.e. action $a_2 = left$) now looks worse than the remainder of $\pi$ (i.e. $a_2 = right$), with CVaR$_{0.1}$'s of $-2$ and 0, respectively; as a result, the fCVaR agent will defect on its original plan $\pi'$.

One way to circumvent this issue with time consistency is to skirt dynamic evaluation altogether and precommit to evaluating risk with respect to only one stage or time-point (the start being most natural) – thus enforcing a commitment $\pi'$ at state A. One way to make this contract is to change the value of $\alpha$ after each transition in the light of the probability of its happening. For instance, the decision in $\pi'$ made at state A to go *left* rather than *right* at state B was based on considering all the outcomes in state B (i.e. the decision was based considering CVaR at $\alpha = 1.0$ rather than $= 0.1$). This dynamic adjustment of $\alpha$ (in this case from 0.1 in state A to 1.0 in state B) prevents time inconsistency by allowing later risk evaluations and decisions all to be coordinated with respect to the single risk preference at the start stage [31].

A second way to deal with time consistency is to evaluate risk dynamically using a series of nested one-step (conditional) risk measures [11, 35]:

$$\rho_{k,N}(R_k, \ldots, R_N) = R_k + \rho_k(R_{k+1} + \rho_{k+1}(R_{k+2} + \cdots + \rho_{N-2}(R_{N-1} + \rho_{N-1}(R_N)) \cdots)), \quad (6)$$

where $R_k$ is the reward (or cost) for time step $k$.

If CVaR$_\alpha$ with the same value of $\alpha$ is used for each conditional risk measure $\rho_k$, this is known as nested or nCVaR$_\alpha$ [36]. Although like fCVaR, it keeps risk preferences fixed across time, it applies the CVaR at each stage to subsequent CVaR evaluations (which themselves are random *scalars*) rather than to the *full distribution* of future random costs or rewards under the policy lower in the tree.

One consequence of this nesting or compounding of risk evaluations, is that nCVaR can sometimes become much more conservative than the other two approaches. To see this, consider panel b in Figure 5. Here, the same decision tree is appended with a set of alternative transitions (depicted in gray) from state B. Now, each possible action is additionally associated with a 10% chance of ending up in state D, which either returns 0 with probability 0.9 or makes a transition to yet another state E with probability 0.1. The outcomes in this state are either $-5$ (the worst possible outcome) or again 0. Importantly, from the perspective of state B, the $-5$ is so remote (having a probability of merely 0.001) that it barely impacts the distribution there (and therefore pCVaR and fCVaR). However, due to the nested structure of nCVaR (at $\alpha = 0.1$), the $-5$ is propagated backwards first to E then to D, and then in fact, all the way back to the top state A. As a consequence, the nCVaR agent would then rather choose $\pi'' = \{a_1 = left, \text{N/A}\}$, for a certain loss of -4.

Of course at $\alpha = 1.0$, all three approaches are equivalent to using the expected value, which is time consistent. As $\alpha$ approaches 0, all three approaches again become equivalent, but to the worst-case risk measure [37], which is also time consistent.

**The three approaches in a gridworld**

As noted, the two-step task is too simple to surface these issues (evaluating the other forms of CVaR leads to equivalent results on the 792 subjects). Thus, we used the simulations in Figure 6 to highlight their differences as the basis of possible future tests.

Consider an agent (or decision maker) who starts in a gridworld in the top left corner and can move either right or left. Exiting the gridworld on the right hand side informally represents a goal and is associated with a +3 reward, while exiting the map on the left hand side represents quitting and a loss of -2. The agent's actions are stochastic with the possibility of moving downwards with some error probability. If the agent falls off the bottom of the gridworld, a substantial loss of -15 is incurred (schematized by the lavapit). The right action has twice the error probability of the left action, meaning that heading towards the goal is riskier than attempting to quit; this leads to interesting differences for the three approaches.

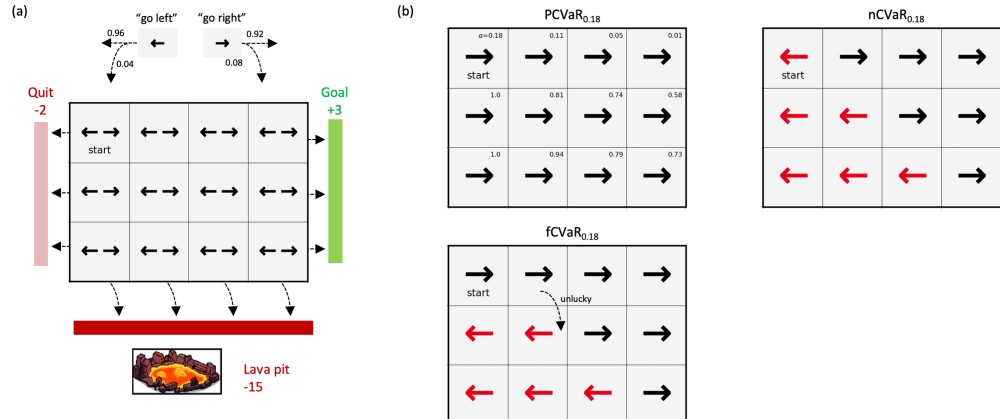

Figure 6: Three approaches to CVaR in a gridworld. (a) The agent starts in the upper left corner and can choose to go left or right in each state. Actions result in stochastic transitions (as depicted at the top). Exiting the map on the left, right, or bottom results in an outcome of -2, +3, -15. (b) The optimal policy is shown for each of three approaches, highlighting their distinct behaviors.

Optimal polices for the pCVaR, fCVaR and nCVaR were calculated using dynamic programming equations (see Appendix B; based on [11, 16]) for various levels of $\alpha$ (for pCVaR, which requires interpolating between different values of alpha, we used 21 log spaced points, following [16]). The optimal policies for each approach to CVaR at a moderate-to-high level of risk (i.e. $\alpha = 0.18$) are depicted in Figure 6b.

For pCVaR (with a start state risk of $\alpha = 0.18$), the optimal policy in every state is to head towards the goal. This is the case even in the bottom row, where rightward actions are much more risky due to the proximity of the lava pit. In fact, if the agent started in the bottom row with the same $\alpha = 0.18$, it would head left in an attempt to quit. However, since the pCVaR approach coordinates risk with respect to the start state and these bottom states are part of the start state's lower tail, the pCVaR agent knows that it is better to make 'riskier' decisions here for the sake of its former self.[3] Indeed, doing so yields a higher start state CVaR than abandoning its plans. In contrast, abandoning the pursuit of the goal is exactly what the fCVaR agent does, as it re-evaluates risk using $\alpha = 0.18$ at every state; i.e. it heads towards the left in rows 2 and 3 after getting knocked off course. Similarly to fCVaR, the nCVaR agent also re-evaluates risk using $\alpha = 0.18$ at every state. However, it chooses to quit from the start due to the nesting of risk (risk evaluated on top of future risk evaluations) – even from the start, the chances of the distal lava pit looms larger than it does for the other two approaches (at the same nominal level of alpha). At lower and higher values of alpha, the behavior of the three approaches more closely aligns, either quitting from the start or pursuing the goal from every state.

## 4   Discussion and related work

We first showed that adopting the modern risk measure CVaR [4] in a form based on ideas from distributional RL [6] suggests that a large minority of healthy volunteer subjects exhibit quite significantly risk-averse behavior. This was true even in a simple two-step task for extremely low stakes and with only rather limited amounts of uncertainty. That this risk aversion masqueraded as enhanced perseverance and slowed learning is a reminder of the complexities of building models of behavior, and an invitation to consider risk in models of other common tasks. These effects of risk in behavior can be complemented by considerations of its effects in the sort of off-line planning that has recently been suggested to model rumination in anxiety disorders, as subjects struggle to find ways of mitigating potential future threats [38]. That distributional RL has recently been suggested as a way of understanding facets of the activity of major neural systems involved in processing affective outcomes [7] offers a highly attractive link to understanding risk aversion in animals (and

---

[3]This increase in risk sensitivity can be seen in the average values of the adjusted alphas in each state; shown in the upper right hand corner of each grid cell.

also tempts us to consider the role of other neuromodulators that have been implicated in representing and processing uncertainty; [39]).

Since pathological effects of uncertainty play a malign role in the effects of many psychiatric conditions [1], it would be of great interest to design tasks, for instance based on the gridworld of section 3, that decompose various of its different facets to help determine which might be responsible. Indeed, this task can be seen as a structurally rich form of a popular method for assessing impulsivity – the balloon analogue risk task (BART; [40]). Many potential variants would be of interest – for instance studying forms of precommitment by allowing subjects to buy a form of 'insurance' at the outset against one or more unlucky downwards transitions. We could modify the task further with intermediate rewards and punishments to examine the observation that individuals only take on greater risk after a loss, if it is not yet realized (in our terms, before reaching one of the two sides of the grid) [41]. Then, we might expect precommitments to be abandoned and risk to be re-evaluated to the degree to which previous outcomes seem irreversible or the current situation sufficiently distinct from the one in which the committment was made. One could examine consistency between subjects' choices in the environment and their willingness to pay to protect themselves ahead of time. Furthermore, richer distributions of possible outcomes and a diverse set of navigation environments, with features that differentiate the three approaches, could be used to reverse engineer uncertainty calculations in the brain given stable risk preferences.

One particular facet of uncertainty that we did not explore is that, in some circumstances, it is possible to collect more information that reduces it to acceptable levels. Indeed, modern risk measures have recently been applied applied in settings with model uncertainty (i.e. POMDPs and Bayes adaptive MDPs, [21–24]). Information gathering, as explored in the sequential information sampling task [42] has been of particular interest in obsessive compulsive disorder, where subjects have been algorithmically modelled as exhibiting less urgency to make up their minds [43]; it would be interesting to model them computationally as being more risk averse. Similarly, if trust is seen as willingness to risk vulnerability to others [44, 45], then risk sensitivity could be an important factor in social pathologies such as borderline personality disorder.

We showed that fCVaR, although intuitive, is not time consistent. The other forms of precommitted pCVaR and nested nCVaR are, which provides them with more attractive formal properties. In fact, one can see fCVaR as living inbetween pCVaR and nCVaR (rather literally, in the problem shown in Figure 6b). There are other potential interpolants – for instance, cases in which the updating of $\alpha$ following lucky or unlucky transitions (which justifies a form of gambler's fallacy; [38]) is incomplete (or perhaps asymmetric). Such incompetent calculations have been suggested as underlying psychiatric conditions themselves [46].

Finally, although we focused on CVaR, there are many other risk measures that satisfy the coherence axioms, and indeed other more stringent conditions. Adding the requirements of comonotonicity and law invariance lead to the class of *distortion risk measures* [47–49] (or equivalently, *spectral risk measures* [50]), which apply a distortion function to cumulative probabilities. This allows them to be linked directly to the *dual theory* of choice [51], therefore inheriting an additional set of rational choice axioms (i.e. the axioms of expected utility theory, with an alternative independence axiom)[52]. CVaR itself is part of this class, and can be used as a basis to construct all other members [34, 49]. Interestingly, the probability weighting function from cumulative prospect theory [53, 54], a popular model in psychology, can be considered a distortion risk measure, even though the full prospect theory adds reference dependence and loss aversion. While prospect theory has been well validated for single decisions, there is substantial opportunity for theoretical and empirical investigations of its realization in sequential decision-making.

## 5   Broader Impact

Coherent risk measures, despite their widespread use in formal applications and favorable theory, have yet to permeate fully psychological and neuroscientific models of decision making, especially for sequential problems. We take early steps in this direction using a widely-recognized and commonly used experimental paradigm, and a psychologically relevant coherent risk measure, CVaR. In doing so, we highlight a key issue that can arise, namely time inconsistency, with a naive application of risk measures to sequential choice, and discuss two alternative (time-consistent) approaches. We expect that bringing awareness to this issue will lead to interesting future discussions and empirical tests

in psychological and neuroscientific communities. We do not anticipate our research to selectively impact some groups at the expense of others. Our study had the limitation of not examining how subjects might compute these various forms of risk sensitivity in a neurobiologically credible manner, or showing that the risk aversion that we inferred from behaviour in the two-step task would generalize to other choices that the subjects might make.

## Acknowledgements

The authors have no competing interests to disclose. We thank Fabian Renz for his contributions to a preliminary analyses of the two-step task and helpful discussions. CG and PD are funded by the Max Planck Society. PD is also funded by the Alexander von Humboldt Foundation.

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
