# A  Further details for the two-step task analyses

## A.1  Parameter recovery

To assess the estimability/identifiability of the model parameters, we performed a parameter recovery analysis (Supplemental Figure 1). New data were generated from the model based on participants' parameters (referred to as 'generative') and the model was re-estimated on these data. The rank correlation between the generative and re-estimated (or 'recovered') parameters was greater than $0.74$ (for all parameters except for the dispersion parameter, $\eta^2$), indicating moderately good estimability/identifiability. The rank correlation for $\eta^2$ was lower ($0.53$), but this is expected, because its identifiability depends on the level of $\alpha$; at $\alpha = 1.0$, for instance, choices do not reflect any outcome uncertainty, making $\eta^2$ unidentifiable. However, the estimability/identifiability of $\eta^2$ was similar to the other parameters (i.e. a rank correlation of $0.7$) in participants with $\alpha < 0.5$ (shown in red). Correlations were all highly significant ($p < 1e-10$), as shown by the $95\%$-confidence intervals around the regression lines.

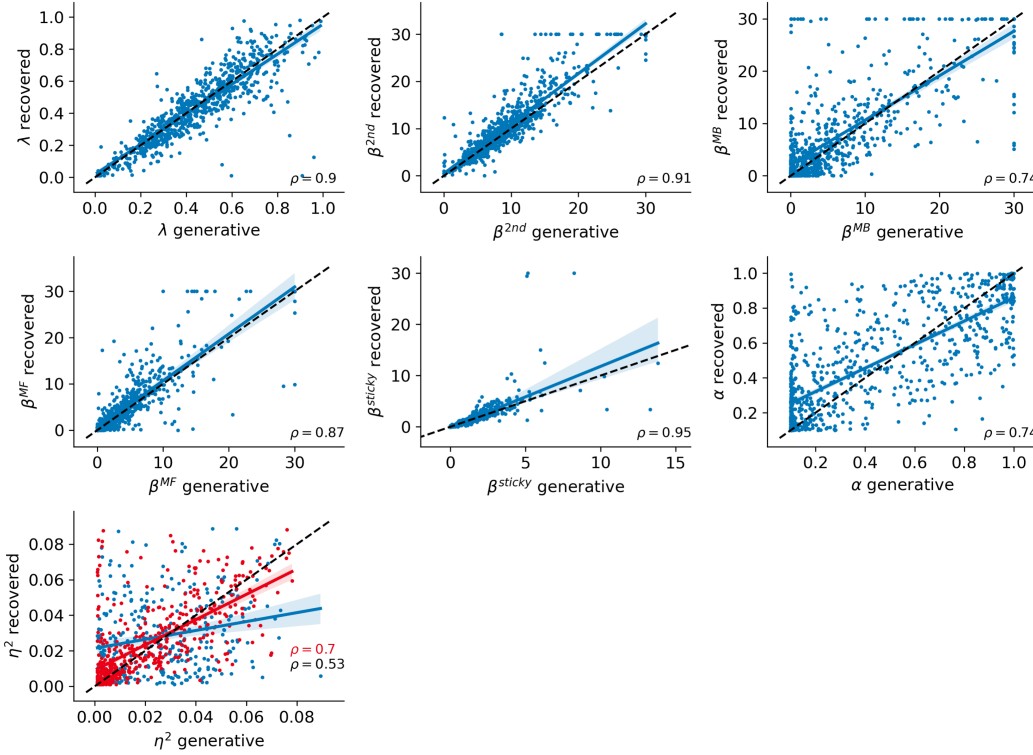

Supplemental Figure 1: Parameter recovery analysis. (See supplemental text for details)

## A.2 Comparison of mean-model to previous model used in two-step task

The mean-model that we used was modified from that in [1] to allow for a more direct comparison to the risk-sensitive model. There are two small differences between our model and theirs. One difference is that the model in [1] uses both rewards and the intermediate values from stage 2 to estimate the values in stage 1 (equation 5). We found this to be unnecessary, as it increased (worsened) the BIC in almost all (>95%) participants. The second difference is one of re-parameterization. The rule that we use to update the mean values (equation 3) keeps the values between $[0, 1]$, while the update rule used in [1] allows them to go above 1. We made this adjustment so that the value estimate had the semantics of a probability estimate. We found that this re-parameterization made very little difference in model fit (average BIC of 371.5 for ours and 373.9 for theirs).

## A.3 Details for the simulation shown in Figure 4

As described in the main text, choices were simulated from the CVaR-model at increasing levels of risk sensitivity ($\alpha \in [1.0, 0.6, 0.3, 0.1]$), in response to a predetermined set of outcomes. The model was forced to choose option A for the first six trials and the outcomes received were $[1, 1, 0, 1, 1, 1]$; from trial seven onward, A always resulted in outcome 0 and B in outcome 1. The other parameters were held constant, with learning rate $\lambda = 0.1$, $\eta^2 = 0.003$, and $\beta^{2nd} = 30$. The initial mean for the value estimates was set to $50\%$ and its initial variance to $0.03$. Since the model was only simulated in response to two options at a single (second-stage) state, parameters that were only relevant to the first-stage decisions were not used (i.e. $\beta^{\text{sticky}} = 0$, $\beta^{MB} = 0$, $\beta^{MF} = 0$).

# B    Three Approaches to CVaR: Dynamic Programming Equations

**Gridworld MDP:**    The gridworld depicted in Figure 6 is treated as a finite horizon undiscounted Markov decision process (MDP). States $s_t$ correspond to locations in the gridworld and actions $a_t$ to a choice of 'going left' or 'going right'. Transitions are stochastic, such that each action results in the desired transition only some percentage of the time; otherwise, the agent transitions downward. The probabilities of an (unlucky) downward transition for the rightward and leftward action were 0.08 and 0.04, respectively. The probability of transitioning from state $s_t$ to $s_{t+1}$ after taking action $a_t$ is denoted by $p(s_{t+1}|s_t, a_t)$. Three states are associated with a non-zero reward or cost: 'Goal', 'Quit' and 'Lavapit' with rewards/costs of $[+3, -2, -15]$, respectively. The reward/cost function is denoted as $r(s_t)$. Visitation to any of these three states ends the episode; i.e., from them, the agent transitions deterministically to a terminal state, which has a reward of 0, and where it remains indefinitely. Otherwise, the episode terminates at horizon $T$.

**Precommitted CVaR (pCVaR)**    The optimal policy for the precommitted approach to CVaR (or pCVaR) was calculated using a modified version of the dynamic programming algorithm in [2]; we adapt their algorithm to the finite horizon setting for an easier comparison with the fixed approach (fCVaR).

Starting at time point $t = T - 1$ and working backwards, the agent calculates a set of Q-values $Q_t(s_t, a_t, \alpha_t)$ for all states $s_t$, actions $a_t$ and risk preferences $\alpha_t$:

$$Q_t(s_t, a_t, \alpha_t) = r(s_t) + \min_{\xi(s_{t+1}) \in \mathcal{U}(\alpha_t)} \sum_{s_{t+1}} p(s_{t+1}|s_t, a_t)\xi(s_{t+1})V_{t+1}(s_{t+1}, \alpha_t\xi(s_{t+1})) \quad (1)$$

The state values $V_t(s_t, \alpha_t)$ are updated by taking a maximum over Q-values,

$$V_t(s_t, \alpha_t) = \max_{a_t} Q_t(s_t, a_t, \alpha_t), \quad (2)$$

or are set equal to 0 at the horizon ($t = T$).

These update equations differ in two notable ways from traditional dynamic programming, which uses a risk-neutral objective. First, the transition probabilities are weighted by $\xi(s_{t+1})$, which are chosen to minimize the next state's value as much as possible, under the constraints that the individual weights are less than $1/\alpha_t$ and that the distorted probabilities still sum to 1. These two conditions are denoted by:

$$\mathcal{U}(\alpha_t) := \left\{ \xi : \xi(s_{t+1}) \in [0, 1/\alpha_t], \sum_{s_{t+1}} p(s_{t+1}|s_t, a_t)\xi(s_{t+1}) = 1 \right\} \quad (3)$$

Second, the value at risk preference $\alpha_t$ at time point $t$ (on the left-hand side of equation 1), is a function of the next states' values at potentially different risk-preference levels, given by $\alpha_{t+1} \leftarrow \xi(s_{t+1})\alpha_t$. The first modification comes from the dual representation of CVaR as a distorted (or $\xi$-weighted) expectation [3][1], and the second modification from the conditional decomposition theorem of CVaR in [4]. The consequence of these modifications is that $V_t(s_t, \alpha_t)$ equals the $\text{CVaR}_{\alpha_t}$ of the future sum of rewards (i.e. the return) starting in state $s_t$ and following the optimal policy from thereon. Since is $\alpha_{t+1}$ is continuous, $V_t$ and $Q_t$ are linearly interpolated using 21 log spaced points for $\alpha_t$, following [2], and thus the values are only approximately equal to the CVaRs. Chow, et al. (2015) interpolate $\alpha V$ and show that this allows them to bound interpolation error in a value-iteration approach; for our finite time horizon simulations, we simply interpolate $V_t$ since we are not iterating the equations until convergence.

The optimal policy $\pi_t(a_t|s_t, \alpha_t)$ is calculated by taking the action associated with the maximum Q-value in each state (i.e. as the argmax in equation 2). For each episode, agent starts at a preferred level of risk $\alpha_0$ (which is set to 0.18 for the simulations in Figure 6) and chooses an action according to $\pi_0(a_0|s_0, \alpha_0)$. However, it then needs to adjust this risk preference depending on the state transitions (or rewards/costs) that occur, for instance, $\alpha_{t+1} \leftarrow \xi(s_{t+1})\alpha_t$ for a particular transition involving $s_{t+1}$. These weights $\xi(s_{t+1})$ used for these adjustments are obtained from equation 1.

---

[1]The dual representation is $\text{CVaR}_\alpha[Z] = \min_{\xi \in \mathcal{U}(\alpha)} E_\xi[Z]$

**Nested CVaR (nCVaR)** The optimal policy for the nested approach was calculated based on [5]. For all states $s_t$ and actions $a_t$ (and for a fixed, preferred risk preference $\bar{\alpha}$), the Q-values are calculated as:

$$Q_{\bar{\alpha},t}(s_t, a_t) = r(s_t) + \min_{\xi(s_{t+1}) \in \mathcal{U}(\bar{\alpha})} \sum_{s_{t+1}} p(s_{t+1}|s_t, a_t) \xi(s_{t+1}) V_{\bar{\alpha},t+1}(s_{t+1}) \tag{4}$$

For each state, the values are calculated as:

$$V_{\bar{\alpha},t}(s_t) = \max_{a_t} Q_{\bar{\alpha},t}(s_t, a_t) \tag{5}$$

The optimal policy $\pi_{\bar{\alpha},t}(a_t|s_t)$ is again calculated by taking the actions that are associated with the maximum Q-value in each state (i.e the argmax in equation 5).

Note that unlike for pCVaR, the values and policy are only represented and updated at a single, fixed risk preference ($\bar{\alpha}$). We denote this using a subscript to emphasize the difference. Also note that the $Q_t$ and $V_t$ no longer correspond to the CVaR of the return, like in the pCVaR case, but rather the nested sum of immediate rewards and future CVaR evaluations, as given in equation 4 in the main text.

**Fixed CVaR (fCVaR)** Calculating the optimal policy for the fixed CVaR approach involves a combination of the precommitted and nested approaches.

At its heart, fCVaR involves a form of distributional RL in which the distribution over Q-values is represented using a set of CVaRs across $\alpha$ levels (again, using 21 possible values) rather than using quantiles [6]. Thus, for all states $s_t$, actions $a_t$ and risk preferences $\alpha_t$, Q-values across $\alpha$ levels are updated using the same equation 1 as the precommitted approach. In terms of distributional reinforcement learning, this corresponds to a distributional back-up.

Diverging from the pCVaR approach, however, and more like nCVaR, the (distributional) value for each state $s_t$ (still represented across multiple $\alpha_t$ values), is set according to the action that maximizes the Q-value for that state at a fixed, preferred level of risk ($\bar{\alpha}$):

$$V_t(s_t, \alpha_t) = Q_t(s_t, a^*_{\bar{\alpha},t}, \alpha_t), \tag{6}$$

where $a^*_{\bar{\alpha},t}$ is chosen to maximize:

$$a^*_{\bar{\alpha},t} = \arg\max_a Q_t(s_t, a, \bar{\alpha}) \tag{7}$$

This alternates between faithfully backing up the distribution and choosing an action at each time step that greedily maximizes the CVaR of the future return at a fixed, preferred alpha level. As we discuss in the main text, this greedy action selection can lead to time inconsistency. Thus similarly to nCVaR, the optimal policy $\pi_{\bar{\alpha},t}(a_t|s_t)$ is only a function of state $s_t$, with the preferred risk preference $\bar{\alpha}$ kept constant across time (yet potentially differing across agents or simulations).

**Depicting the optimal policies:** Given the finite-horizon nature of the problem, the optimal policy for each state could potentially differ across time. Furthermore for pCVaR, the optimal policy can also differ depending on the risk-preference $\alpha_t$, which is adjusted after the start of the episode depending on the transitions that occur. Therefore, we used simulations to explore how the optimal policy unfolded across time for each agent. The agents started in state $s_0$ (upper left hand corner) and chose actions until they reached either the 'Quit', 'Goal', or 'Lavapit' state (this always occurred before $t = 10$). In all of the simulations (20,000 for each agent), the action taken in each state was the same, regardless of the time point. This was true even for the pCVaR agent, which sometimes visited different alpha levels within a state (depending on how it got there). The actions from these simulations were plotted in Figure 6. The optimal policies for other levels of $\alpha$ are further explored in Supplemental Figure 2.

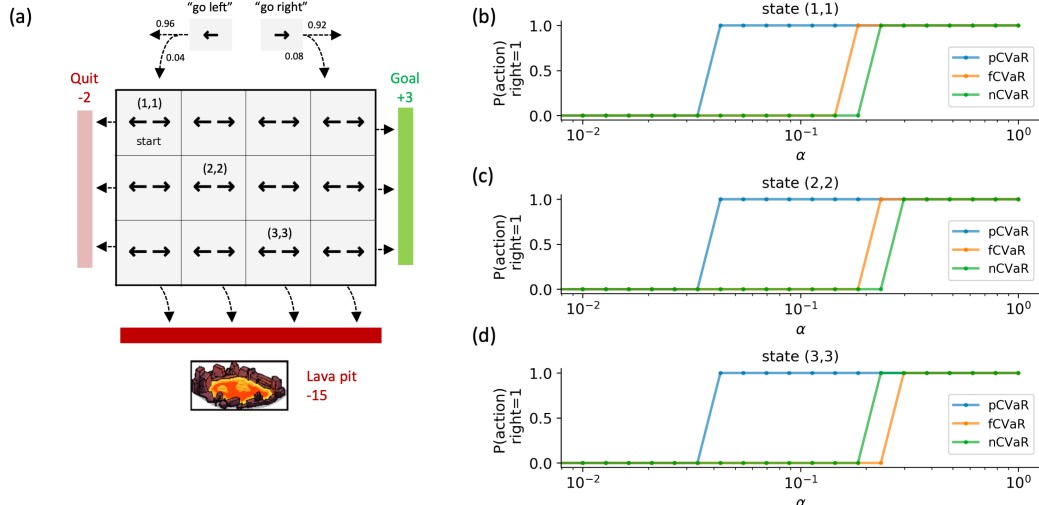

Supplemental Figure 2: Optimal policies at other $\alpha$ levels. (a) Shows the gridworld MDP (also shown in Figure 6 in the main text). (b-d) The optimal policy is plotted as a function of $\alpha$ for pCVaR, fCVaR and nCVaR for three states: $(1,1)$, $(2,2)$, $(3,3)$, whose locations are labeled in panel a. The optimal policy was examined by running simulations starting in state $(1,1)$. The probability of taking a rightward action in simulation is plotted (this probability was either 0 or 1, as the agents always took the same action in each state). If a state was not visited during any of the simulations, the policy for the agent starting in that state is plotted (e.g. for states $(2,2)$ and $(3,3)$ at low alphas, where the agent decides to quit from the start). All three methods choose to go right towards the 'Goal' state at high levels of $\alpha$ and left towards the 'Quit' state at low levels of $\alpha$. The three methods differ, however, in when they choose to make this switch. pCVaR chooses to go right towards the goal at much lower levels of $\alpha$ than the other two methods in all three states. nCVaR requires the highest levels of $\alpha$ before it chooses right in states $(1,1)$ and $(2,2)$, reflecting its more conservative risk evaluation of distal threats (i.e. the lavapit). However, for state $(3,3)$, which is adjacent to the lavapit, it chooses to go right at a lower $\alpha$ level than fCVaR (but still a higher level than pCVaR). The 21 interpolation points used for $\alpha$ are shown as dots plotted on the lines.