# OpenReview forum: "Two steps to risk sensitivity"
_NeurIPS.cc/2021/Conference — NeurIPS 2021 Spotlight_

### Official Review · Reviewer_xqSF · 2021-07-15

**Rating:** 6
**Confidence:** 2

**Summary:**

This study proposes how we can integrate distributional RL and CVaR to account for seemingly irrational human behavior. CVaR is widely accepted in finance but not yet in other fields provides a new way to measure the level of subjective risk. Recent findings suggest that the parameter associated with CVaR may well capture the individual differences in risk sensitivity.  This study shows that the model combining the distributional RL and CVaR indeed explain better human choices in two-step task. Furthermore, this study provides a potential solution of how this method can be extended in multistep decisions in a more complicated task. While this model is better predicts the actual choices of human subjects, many parts of this model have not yet fully tested, as the author pointed out, because the data was acquired from too simple a two-step task. Despite this limitation, this might be appreciated by broad research fields (e.g. from investigating psychiatric diseases to designing robots or self-driving cars). I have few comments to make this study to be appreciated more by other filed of researchers.

**Limitations And Societal Impact:**

yes.

**Main Review:**

1. What is assumed to be known or unknown to participants? What is the condition that this model works? It seems that participants should know the task structure such as transition but they might not know the true probability of transition between states. However, it is rare that the graphical configuration is given to participants. Participants should infer the latent task structure, especially in a two- or n-step task. Having a good representation of the transition structure would be more difficult when the task is getting complicated. It might be necessary to mention what information is given to participants or pre-trained. If participants use a hybrid model and have a tendency to choose the previous option if they get the rewards (such win-stay-lose-switch strategy is different from stickiness or perseveration), how well alpha can be estimated?

2. Is ρ={ρk, ... ρN-1} assumed to have the same value per subject? Participants might be more sensitive to the risk of the recently visited state. However, this temporal decay might already be achieved by the distribution - updating the distribution of unvisited state close to the uniform distribution -.  Can the authors confirm if this is correct?

3. Since the models are not competing to explain variances of the data, I wonder how much fCVaR and nCvaR can be discriminated when comparing the model fits. Can it be distinguished which algorithm a participant uses, or they are introduced as two methods to model the behavior but give similar results? For example, in the grid example, these two models predict different behavior under the same alpha. However, can they have the same prediction but with different alpha? If pseudo data is generated by one of them, how much does the other model closely estimate the alpha? Do they give similar alpha values or does one tends to estimate alpha higher or lower than the other?

4. What is the minimum number of trials required to estimate the stable alpha value and how it changes when the task gets complicated? What are factors should considered to compute the minimum number of trials? For example, does the number of minimum responses to estimate accurate parameters differ by the alpha values?


**Time Spent Reviewing:**

4

---

> ### Author Response · Authors · 2021-08-06
> **Response to Reviewer xqSF**
>
> Dear Reviewer xqSF,
>
> Thank you very much for your constructive comments. We will respond to each in turn.
>
> (1)
> We will add as much information about the experimental set-up (e.g. task, instructions, etc.) as possible while keeping within the (now extended) page limits. To answer your questions here: Participants were given 40 trials of practice before the actual experiment, during which they learned about the general structure of the task including the 70%/30% transition probabilities. This structure was not presented to participants explicitly, but modeling participants’ choices as if they had learned about it has provided a good fit to behavior in many previous studies. Furthermore, it was fixed throughout the experiment, whereas the participants were instructed about the drifting reward probabilities, and explicitly experienced these changes. After the practice phase, participants were tested about their knowledge of the task and excluded if they failed this test.
>
> In the light of your comment, we conducted two analyses to investigate the potential impact of win-stay-lose-shift heuristics. We appreciate this opportunity to illustrate the robustness of our methods.
>
> First, we simulated data from a model that selected choices using a win-stay-lose-shift heuristic and we fit this data with our full choice model to see whether the model would erroneously estimate risk-sensitivity. Repeating this 100 times, we found that 91 simulated datasets (participants) had estimates of CVaR alpha >=0.99, thus correctly estimating no risk sensitivity in these simulated datasets (note that CVaR alpha=1 is completely risk neutral). This contrasts starkly with the real data, in which 57% of the participants had CVaR alpha estimates below the lowest value (alpha=0.44) obtained in this simulation. Additionally, the estimated learning rates in this simulation were all greater than 0.55, which is expected, because the win-stay-lose-shift strategy is a special case of the model-free component of our model, i.e. when it has a high learning rate. Please also note that our original parameter recovery analysis in the appendix included many simulations with high learning rates, and so the fact that we can recover the CVaR alpha well in those cases, implies that a win-stay-lose-shift strategy will not be affecting our conclusions.
>
> Second, in response to your suggestion to investigate a hybrid model, we replaced the preservation/sticky component in our model with a win-stay-lose-shift component; everything else in the model was the same, so choices were partially determined by the win-stay-lose-shift heuristic and partially determined by the (risk-sensitive) MF/MB components. Using this hybrid model to both simulate data and estimate parameters, we ran a parameter recovery analysis with 100 simulated datasets; this analysis was conducted in the same way as the one in the supplements. In this new analysis, we obtained a moderately good recoverability, with a rank-correlation of rho=0.66 between the generative and recovered CVaR alpha, indicating that the presence of a win-stay-lose-shift strategy does not severely impact the ability to estimate risk sensitivity.
>
> (2)
> In the nested CVaR model, we currently assume that participants would have the same value for alpha (risk preference) at each future time point. The more general nested formulation for time-consistent dynamic risk measures [11] does allow for different one-step conditional risk measures. However, we expect that it would be very difficult to estimate different alpha values at each time point. It might be possible to estimate a parametric decrease (or increase) in alpha; but, as you suggest, a decay towards uniform distributions would also produce shifts in the tails of the distributions and therefore shifts in risk preferences for frequently/infrequently visited states. Teasing these apart would require careful task design.
>
> (3)
> We chose the example gridworld in Figure 6 to illustrate primarily the differences between pCVaR and the other two versions (fCVaR and nCVaR), because we think that this distinction has the most interesting psychological and neuroscientific implications. You are right, however, that this example does not distinguish fCVaR and nCVaR very well across all alpha levels; and indeed, at different alpha levels, behavior (for pseudo data or real data) under these two versions would be confused. This is unsurprising, because even at alpha=0.18 in Figure 6, they only differ in the choice made at one state (state=(1,1)).
>
> However, there are other gridworlds (we give an example below) that better disentangle fCVaR and nCVaR, even if they slightly suppress the differences to pCVaR. For instance, using the same gridworld but changing the error probabilities to 0.05 and 0.01 (for going right and left, respectively) leads to different behavior for fCVaR and nCVaR at other alpha values, namely 0.1 and 0.15. At alpha=0.1, the pattern seen in Figure 6 -- where nCVaR chooses to exit the map at the starting location whereas fCVaR does not -- reverses. At this same alpha level, there are also two other grid locations where fCVaR and nCVaR would choose other actions. This is shown in the reproduced gridworlds below where the action in each grid location are depicted as 1=”go right” and 0=”go left”:
>
> alpha=0.1
>
> nCVaR
>
> [1. 1. 1. 1.]
>
> [0. 0. 0. 1.]
>
> [0. 0. 0. 0.]
>
> fCVaR
>
> [0. 1. 1. 1.]
>
> [0. 0. 1. 1.]
>
> [0. 0. 0. 1.]
>
> And at alpha=0.15, both would choose to pursue the goal from the start, but they would make different choices in the second row. This can be seen below:
>
> alpha=0.15
>
> nCVaR
>
>  [1. 1. 1. 1.]
>
>  [0. 0. 0. 1.]
>
>  [0. 0. 0. 1.]
>
> fCVaR
>
> [1. 1. 1. 1.]
>
> [0. 1. 1. 1.]
>
> [0. 0. 0. 1.]
>
>
> Furthermore, while individual gridworlds, such as the one above and the one in Figure 6, may only differentiate the algorithms over a somewhat narrow range of alpha values, combining multiple gridworlds would provide better estimates for both a participant’s algorithm and their alpha value. This, or even an adaptive staircase design, would be a standard procedure in a psychological experiment aimed at inferring the nature of risk sensitivity in human subjects.
>
> To investigate this, we created 5 gridworlds with different error probabilities (from 0.04 to 0.2) and costs (from -15 to -25 for the lava pit) and compared the policies for the three algorithms across a range of different alpha values. Although different algorithms at different alphas sometimes had the same policy in one or two gridworlds (e.g. fCVaR at alpha=0.2 and pCVaR at alpha=0.3), they almost never had the same policy in all 5 gridworlds. This was true for all values of alpha between 0.05 and 0.61, except for nCVaR alpha=0.18 which happened to have the same set of policies as fCVaR alpha=0.14. For values of alpha >0.61, there were more confusions, but this is expected since as alpha approaches 1 all three approaches are equivalent to the expected value. In a real experiment, there would additionally be noise in participants’ choices, but there would also be the opportunity to run dozens of different gridworlds (or even more complicated navigation problems).
>
> (4)
> These are important considerations. For the two-step task, our recovery simulations suggest that we can get stable estimations of the value of alpha within 200 trials -- and this is not greatly influenced by the value of alpha itself. As evidenced by the data, this is amply possible for online studies.
>
> We have not yet optimized the design of the gridworld task for use in human subjects, and so we cannot make a reliable guess about its estimation properties. However, it is certainly the case that telling the different algorithms apart will become increasingly impossible as alpha gets near to 0 or 1, since they all become equivalent. If subjects used a mixture of the strategies, such as incomplete adjustment for un/lucky transitions, this would make estimation yet harder.

---

### Official Review · Reviewer_nvvN · 2021-07-16

**Rating:** 8
**Confidence:** 5

**Summary:**

This paper suggests a new model of behaviour based on conditional-value at risk principle that can provide interesting alternative interpretations to the "two step" task used in many "model-free" vs. "model-based" RL studies. Although the scope of the framework is somewhat narrow (albeit with a variety of modifications to address time consistency), the paper provides a number of interesting explanations to the data that should be considered for the sake of accurate interpretation of computational model-based analyses. The analyses are also very detailed and interesting although I would have preferred to see more focus on empirical data as opposed to theoretical considerations. Overall a strong paper for NeurIPS.

**Limitations And Societal Impact:**

Yes. Perhaps should discuss about other relevant models outside of CVaR framework a bit more.

**Main Review:**

Originality: Although the CVaR method is not new, its application to this kind of problem and analysis of all the consequences is original. The gridworld task used to illustrate differences between different model variations is novel (in this context).

Quality: The study seems of really high quality with a good review of literature, critical analysis of relevant issues (notably likely misinterpretation of inferred parameters in the discussed tasks/studies), and detailed analysis of consequences of the proposed model, including time sensitivity, an important even a somewhat narrow issue. I think readers can be convinced of relevance and importance of this kind of model extension.

Clarity: Overall the paper is quite clear, even though Fig. 5 and its explanation is a bit confusing. The first paragraph of results refers to Fig. 3, not 2. The figures are nice and neat.

Significance: Although the proposed model extension is somewhat narrow (and there might be other more elegant ways to achieve the same outcomes, e.g. incorporating prospect theory as opposed to simply considering mean of the lower range of outcomes, which could be discussed more in detail), its consequences to interpretation of model parameter results are quite striking. The discussion is very interesting, putting the results in a good perspective. It's nice that the authors go beyond the simple two-step task boundaries but even the gridworld task is fairly simple (compared to what happens in the real world).



**Time Spent Reviewing:**

4

---

> ### Author Response · Authors · 2021-08-06
> **Response to Reviewer nvvN**
>
> Dear Reviewer nvvN,
>
> Thank you very much for your comments and helpful suggestions for how the paper can be improved. We will clarify the explanation for time-consistency and Figure 5, and fix the erroneous reference to Fig. 3.
>
> As you suggest, we will also add more to the discussion about other approaches to modeling risk, especially prospect theory. In the future, we are interested in exploring how other models make predictions for the two-step task and for (slightly) more complicated sequential problems, such as the gridworlds we sketched out.
>
> We chose CVaR for this paper for a few reasons: It is simple (adding only a single estimable parameter), recent work has argued that it captures the sort of risk preferences that are particularly relevant to psychopathology [40], and it is part of the class of distortion risk measures (a subset of coherent risk measures), which justifies it with respect to both financially-relevant axioms as well as the axioms of the economic dual theories of choice (Yaari 1987; Quiggin 1982). Interestingly, the probability weighting function from cumulative prospect theory can be considered a distortion risk measure, even though the full prospect theory adds reference dependence and loss aversion.
>
> It is also interesting that while prospect theory has been well validated for single decisions, there is much to be explored for it, both theoretically and empirically, for sequential decisions. As far as we know, the issue of time inconsistency has not been addressed, nor with it precommitted/nested/fixed preferences; and neither has the possibility that the dynamics of the reference point should be tied to the good- or bad-fortune experienced in sequential transitions, as can be necessary for CVaR.

---

> > ### Comment · Reviewer_nvvN · 2021-09-03
> > **my response**
> >
> > I thank the authors for their efforts to improve the paper in line with my suggestions.
> > I would also suggest them to focus a bit more on interpretation of parameter values, especially with regard to differences from learning rate-based interpretation of the experimental results. I see that, rather than the technical part (which can be improved in line with suggestions of other reviewers), as the strongest/most consequential part of the paper why I think it is a strong paper worthy of acceptance at NeurIPS, capable of stimulating discussions in the computational cognitive neuroscience community. In addition, it may help discussing model-based analysis literature beyond the "two step" task and risk where there also have been some pretty fundamental discrepancies between papers trying to attribute all/most of their effects to the learning rates as opposed to those using models with more flexible parameters (e.g. exploration-exploitation balance and discounting factor in reinforcement learning), where the significant differences usually get attributed to those other parameters.

---

### Official Review · Reviewer_zeBh · 2021-07-16

**Rating:** 4
**Confidence:** 3

**Summary:**

The authors adopt a risk measurement to a well-studied two-step RL task and show that many humans are risk-averse. This provides an alternative explanation (risk aversion) for previously assumed stickiness or perseveration. The authors also discussed time consistency issues with their models and provided a few better alternative models and a better experimental design to test it.

**Limitations And Societal Impact:**

Yes

**Main Review:**

The task and data are not new, they are from previous publications. The method or the combination of task and method is somewhat new -- applying an existing measurement to a different setting (sequential decision-making).

The paper is very interesting and interdisciplinary, but overall I’m not very positive about the paper. The model comparison results indicate that the new risk-sensitive model might on average be as good as the mean model. The averaged BIC is not reported, nor is the percentage of subjects that are better by the risk-sensitive model. If what I suspected is true (two models are almost equally good), the paper contributes an alternative explanation for stickiness/perseverance, which can be interesting too. However, the results that uncertainty/risk affects decision-making(perseverance) or modulates learning rate are not very novel (e.g. Behrens et al, 2007), and both explanations (risk-aversion vs. perseverance) seem descriptive to me. The authors extensively discuss their model on time consistency and future directions on this line of research, which are very interesting and can be fruitful. However, they are a work in progress, which usually does not warrant an acceptance at neurips.

The paper is not hard to understand, but it is a bit hard to follow. I guess one reason is that the paper does not have the common structure that a ML paper has, which is totally fine. It’s great that the authors provide explanations, but then the most critical definition/information is kind of hidden in a dense paragraph with information that can go into intro&method&discussion. I think this might come from a different writing style of a science paper vs. a more technical paper. I think one quick fix would be to leave them (e.g. VaR) stand-alone, like equation (1). Some “dry” math definitions before explanation/intuition can help. Reducing the number of footnotes can also help.
Time consistency can be better introduced, especially as it is one of the key concepts discussed in this paper. If possible, a real-life example might serve the best. A lot of concepts from psychology/economics can be very new to the majority of neurips readers. It would be impossible to explain them all, but maybe the very key ones can be further explained without assuming domain knowledge from the readers.

Some additional questions and comments:
1. Is the two-step task structure necessary/critical for this paper?
2. What are the distributions below the second stage option in figure 2(a)?
3. It seems that the reward probability has a Markov structure to me. What is the generative model of figure 1(b)? Is this Markov structure considered for MB learning?
4. It would be good to briefly mention how the mean-model differs from the one used in [3]. And if they are quite different, it might be good to include the model in [3] as a baseline model.
5. It’s good to include the percentage of subjects that are better fit by the risk-sensitive model.
6. Can the model capture/generate risk-seeking behaviour?
7. I’m not sure if this data is available, but it would be interesting to correlate alpha in the model with any risk attitude scale.
8. I’m not a domain expert, but it seems to me that the risk in this paper might contain both risk and ambiguity, since the subjects also need to learn the reward probability. Please ignore this if I’m wrong.




**Time Spent Reviewing:**

4

---

> ### Author Response · Authors · 2021-08-06
> **Response to Reviewer ZeBh**
>
> Dear Reviewer zeBh,
>
> Thank you very much for taking the time to review our work. We will address the main review first and then answer specific questions.
>
> Response to main review.
>
> First to provide some relevant information: the CVaR-model provides a better fit in 40.2% of the participants. Due to the large BIC difference for some participants in favor of the CVaR-model, this results in a slightly yet significantly better overall fit for the CVaR model (mean BIC=368.8) compared to the mean model (mean BIC=371.5; diff BIC -2.67,  t=-3.54, p=0.004, paired t-test). We will include these statistics in the revised paper. Since this difference is modest, we agree that it is reasonable to conclude that the models are almost as good on average. However, the large individual differences in BIC imply that modeling risk sensitivity is important in many participants. In fact, we were surprised that so many participants in a general panel playing for small sums in a task not optimized for this characteristic showed substantial risk aversion. We suggest that this speaks in favor of the robustness of our findings.
>
> As pointed out, a core contribution of the paper is to segregate components of preservation. Typically, a unitary form of perseveration is added to models as a heuristic to account numerically for the propensity of some participants to repeat choices, separate from the reinforcement histories of these choices. By contrast, we provide a computational justification for such repetition -- one that suggests relationships to different psychopathological traits (e.g. anxiety versus obsessive compulsive thoughts), and implicates different neural mechanisms. Furthermore, we suggest that just as accounting for risk allows for a less-corrupted estimation of learning rates, it also allows for a less-corrupted estimation of underlying perseveration. Evidence for this is that the model that includes risk sensitivity and perseveration fits significantly better on average than models that contain only one or the other. We will flag this in the revision.
>
> You are quite right that the impact of uncertainty on learning/choice has been studied extensively before. However, our results are largely orthogonal to this previous work. Behrens et al. (2007), for example, manipulated the volatility of the environment to investigate how participants’ estimations of uncertainty (centered on volatility) can impact actual learning rates, as would arise in normative models such as the Kalman filter. In the two step task, environmental volatility is fixed; and our finding is that participants’ preferences concerning uncertainty corrupt the inferences that experimenters have been making about model-free learning and learning rates. This is quite different.
>
> We are very grateful for your comments regarding organization. We will certainly make more stand-alone equations, e.g. VaR and the equation for 2nd stage choices (line 102). We can also move some of the interpretation/background away from the mathematical descriptions. For instance, moving the “Human reasoning is often modelled using…” (line 53) away from the definitions of VaR and CVaR. We will also move the important footnotes to the main text to reduce clutter.
>
> Specific points.
>
> (1) Is the two-step task structure necessary/critical for this paper?
>
> No, it is not strictly necessary. Other Markov decision processes would have also worked. However, the two-step task is ideal, because it has been so widely used in neuroscience and psychology -- with data from 100’s of participants being readily available. Moreover, the fact that it was not designed to investigate differences in risk preferences makes our findings regarding risk sensitivity and perseveration more salient.
>
> (2) What are the distributions below the second stage option in figure 2(a)?
>
> These are examples of the type of distributions that the model, through equations (2), assumes that participants are estimating for each of the 4 choice options. They represent uncertainty about the reward probabilities themselves, and are thus closer to ambiguity (unknown probabilities or second order uncertainty) than pure risk (known probabilities). This relates directly to the next question [which we take out of order]
>
> (8) I’m not a domain expert, but it seems to me that the risk in this paper might contain both risk and ambiguity, since the subjects also need to learn the reward probability. Please ignore this if I’m wrong.
>
> You are certainly correct. We are unfortunately caught between two communities -- one which likes to keep risk and ambiguity separate and another which applies risk measures, such as CVaR, to both types of uncertainty; we cite a few machine learning examples from the latter in the discussion (starting at line 266). We chose to use the terminology of ‘risk sensitivity’ so that it was consistent with calling CVaR a risk measure. However, we did not want to make a strong claim about preferences associated with one type of uncertainty over the other. Indeed, the two-step task is not well equipped to do so. To remedy this, we will mention ambiguity at the beginning of the paper to ward off confusion and make it clear the CVaR is being applied to estimated distributions over reward probabilities.
>
> (3) It seems that the reward probability has a Markov structure to me. What is the generative model of figure 1(b)? Is this Markov structure considered for MB learning?
>
> The drifting reward probabilities are indeed Markov and generated as Gaussian random walks. This structure is not directly considered in the model for learning. As in the previous work [3], we assumed that participants estimate these drifting probabilities using a model-free prediction-error update rule (for the mean; variance estimation was not previously considered).
>
> (4)  It would be good to briefly mention how the mean-model differs from the one used in [3]. And if they are quite different, it might be good to include the model in [3] as a baseline model.
>
> We agree. There are two small differences, which we describe here and will report more clearly in the paper. One difference, currently mentioned in footnote 3, is that the model in [3] uses both rewards and the intermediate values from stage 2 to estimate the values in stage 1. We found this to be unnecessary, as it increased (worsened) the BIC in almost all (95%) participants. The second difference is one of re-parameterization. The rule that we use to update the mean values (equation 2) keeps the values between [0,1], while the update rule used in [3] allows them to go above 1. We made this adjustment so that the value estimate had the semantics of a probability estimate. We found that this re-parameterization made very little difference in model fit (avg BIC of 371.5 for ours and 373.9 for theirs).
>
> (5) It’s good to include the percentage of subjects that are better fit by the risk-sensitive model.
>
> We will include this. It is 40.2%, as mentioned above.
>
> (6) Can the model capture/generate risk-seeking behaviour?
>
> We fit an alternative version of the model which allows the alpha percentage to represent the upper rather than the lower tail, in order to capture risk-seeking behavior. This had a worse fit on average compared to the risk averse model (BIC of 380 versus 368.8), and we omitted it due to space constraints.
>
> (7) I’m not sure if this data is available, but it would be interesting to correlate alpha in the model with any risk attitude scale.
>
> Risk attitude scale data was unfortunately not available, but it would be excellent to validate our model using a scale like this.

---

> > ### Comment · Reviewer_zeBh · 2021-09-01
> > **Reply for rebuttal**
> >
> > I appreciate the authors' effort in responding to my concerns and other reviewers' concern. After reading all the reviews and responses, I'm still overall not very positive about accepting the paper.
> >
> > My current major concern is model goodness-of-fit. As the authors acknowledged, the proposed CVaR-model seems to fit as good as the previous model. This might be fine if authors can provide other model comparisons showing that the new model can capture some important effects that the previous model can not. Posterior predictive check is rather important. My apology for not pointing this out in the initial reviews, as I did not anticipate that the new model is only marginally better than the old one. Along this line of concern, I find that the experiment is not designed for testing this model can be a minus. Doing computational modeling is to understand some effects or behavior we observe, for example, why it is that way or how it is generated. So the paper can benefit from some "model-free" measurements that show consistent effects with what the model predicts (here model-free means some behavioral effects that can be observed/measured without using any modeling). It's important to know that the better model is indeed better at capturing some interesting behavior rather than some "noise" in some subjects. Without fully convincing the readers that the new proposed model is actually a better model, the following implication and discussion are weaken.
> >
> > The idea of this paper is indeed interesting. I do see potentials in this line of research. With a better experimental design or a more careful model comparison analysis, this can be an impactful work. However, unfortunately, I do not see the current version to be a strong submission for NeurIPS.

---

### Official Review · Reviewer_DTEP · 2021-07-21

**Rating:** 5
**Confidence:** 4

**Summary:**

The paper considers using CVaR in human behavior modeling of a two-stage task. In particular, the paper uses CVaR in modeling the choice probability in the two-stage decision-making setting. To work with the time-inconsistency of CVaR, the authors have tried three approaches, namely the fixed, precommitted, and nested CVaR. Using the choice model in analyzing a dataset, the authors show that some participants exhibit risk-averse behaviors.

**Limitations And Societal Impact:**

The authors have adequately addressed the limitations of their work.

**Main Review:**

The paper is clearly written and the motivation of the paper is well-explained. However, it is unclear what the main contribution of the paper is. The current manuscript is an application of existing methods for choice model estimation and constructing time-consistent risk measures from time-inconsistent risk measures, e.g., CVaR. The main novelty of the paper comes from the proposed choice model. However, more explanation and experiments (especially on the sensitivity of the modeling assumption) need to be provided in order to show that such a choice model should be used.

**Time Spent Reviewing:**

2 hours

---

> ### Author Response · Authors · 2021-08-06
> **Response to Reviewer DTEP**
>
> Dear Reviewer DTEP,
>
> Thank you very much for your review of our paper.
>
> We intended the main contributions of this paper to be (a) to show quantitatively that risk sensitivity, which has important psychiatric and societal relevance, plays an important, and previously unheralded, part in performance in a task that has been used in dozens of studies in neuroscience and psychology -- something that is true even though the task was not designed to elicit interesting risk attitudes; (b) that this offers a novel explanation of perseveration -- which has hitherto mostly been just a heuristic used to repair models of behavior; (c) to show that the most straightforward and conventional approach of taking into account risk in distributional RL leads to time-inconsistent choices; and (d) to show how to repair this in ways that have theoretical backing from the finance/ML literature.
>
> In terms of modeling assumptions: we suggest that our account has strong support: it fits behavior well in a large percentage (~40%) of participants, and in some, very much better than the model typically used for this task. We use simulations to show in the supplements that the parameters are largely recoverable and also compare our model with alternatives, such as a beta-binomial version; these analyses were buried in the footnotes, but we will include them in the main text. We sought to make only small changes to the model used previously for the two-step task so that we could offer clear and compelling evidence that overlooking risk sensitivity when modeling choice can have a large impact on the interpretation of the results (e.g. perseveration versus risk preference) and on the estimation of other parameters (e.g. learning rates).
>
> We would also suggest that our work is significant because of its interdisciplinarity. As attested by another reviewer, while many readers will be familiar with some concepts (e.g. CVaR, time-consistent risk measures, etc.) and other readers other concepts (e.g. the two-step task, behavioral models of choice and reward learning, etc.), we expect that few readers will be experts in both domains and so their combination here is novel and interesting. For instance, many neuroscientists and psychologists might find it surprising that the most intuitive (‘fixed’) way of adding risk sensitivity to distributional RL leads to time-inconsistent choices. Conversely, many machine learning researchers might be surprised by the predictions for human decision-making associated with the demands of time-consistency, and pleased by the formal contributions they can make to the understanding of the psychiatry of risk-sensitivity, such as anxiety disorders.

---

### Decision · Program_Chairs · 2021-09-27

**Decision:**

Accept (Spotlight)

**Comment:**

There was extensive discussion on this paper.  Perhaps the largest concern among reviewers was the significance of the model fit though I believe the author response on this issue is satisfactory.  While this paper ended up as a split decision among reviewers, some strong support and comments like the following particularly stood out to me: "[the paper shows] how a different kind of model can provide a substantially different interpretation than numerous other studies" and "[the paper could] encourage other researchers to study how the brain implements distributional RL to make everyday decisions".  Given that this paper proposes and defends a novel perspective in an interdisciplinary area highly germane to current NeurIPS discourse (also with high potential impact), I believe the paper has passed the bar for NeurIPS acceptance.